# Antibacterial Activity of Jelleine-I, a Peptide Isolated from Royal Jelly of *Apis mellifera*, Against Colistin-Resistant *Klebsiella pneumoniae*

**DOI:** 10.3390/toxins17070325

**Published:** 2025-06-25

**Authors:** William Gustavo Lima, Rayssa Maria Rodrigues Laia, Julio Cesar Moreira Brito, Daniel Augusto Guedes Reis Michel, Rodrigo Moreira Verly, Jarbas Magalhães Resende, Maria Elena de Lima

**Affiliations:** 1Faculdade Santa Casa de Belo Horizonte, Belo Horizonte 30110-005, Brazil; rayssalaia4@gmail.com (R.M.R.L.); marielena@faculdadesantacasabh.edu.br (M.E.d.L.); 2Fundação Ezequiel Dias (FUNED), Belo Horizonte 30510-010, Brazil; julio.brito@funed.mg.gov.br; 3Departamento de Química, Faculdade de Ciências Exatas, Universidade Federal dos Vales do Jequitinhonha e Mucuri, Diamantina 39100-000, Brazil; daniel.michel@ufvjm.edu.br (D.A.G.R.M.); verly.rodrigo@gmail.com (R.M.V.); 4Departamento de Química, Instituto de Ciências Exatas, Universidade Federal de Minas Gerais, Belo Horizonte 31270-901, Brazil; jarbasufmg@hotmail.com; 5Programa de Pós-Graduação em Inovação Farmacêutica, Universidade Federal de Minas Gerais, Belo Horizonte 31270-901, Brazil

**Keywords:** colistin-resistant *Klebsiella pneumoniae*, Jelleine-I, pandrug-resistant bacteria, antimicrobial peptide, antimicrobial pharmacology, drug development

## Abstract

*Klebsiella pneumoniae* can acquire resistance mechanisms to colistin and present a pan-resistant phenotype. Therefore, new alternative agents are imperative to control this pathogen, and the peptide Jelleine-I stands out as a promising prototype. Here, the antibacterial activity of Jelleine-I against clinical isolates of colistin-resistant *K. pneumoniae* (CRKP) was investigated. Antimicrobial activity was assessed by determining the minimum inhibitory concentration (MIC), minimum bactericidal concentration (MBC) and time kill-curve assay. The release of 260 nm-absorbing materials (DNA/RNA) and the release of proteins were used in the lysis assay. Anti-biofilm activity was studied in microplates. In vivo activity was determined by the lethality assay using *Tenebrio molitor* larvae. The results show that the MIC of Jelleine-I ranged from 16 to 128 µM and the MBC was on average 128 µM. Jelleine-I at 200 µM killed all CRKP cells in suspension (10^6^ colony-forming units (CFU)/mL) after 150 min of incubation. Jelleine-I acts on the CRKP cell membrane inducing lysis. Biomass and viability of CRKP-induced biofilms are reduced after treatment with Jelleine-I, and the use of this peptide in *T. molitor* larvae infected with CRKP reduces lethality and improves overall larval health. In conclusion, Jelleine-I is a potential prototype for the development of new antimicrobial agents.

## 1. Introduction

*Klebsiella pneumoniae* is a glucose-fermenting, gram-negative, lysine decarboxylase-positive, ornithine decarboxylase-negative, encapsulated, immotile, facultatively anaerobic bacterium [1,2]. This species was originally isolated by Edwin Klebs in 1875 from the airways of a patient dying of pneumonia. Later, *K. pneumoniae* was also described by Carl Friedländer in 1882, which led to it being called Friedlünder’s bacillus for a time [3]. Currently, *K. pneumoniae* is classified into the Enterobacteriales family and is recognized as one of the most opportunistic pathogens involved in healthcare-associated infections (HAI), causing pneumonia, meningitis, bloodstream infections, and urinary tract infections [4]. In fact, *K. pneumoniae* is responsible for approximately 11.8% of all hospital-acquired pneumonia and nearly 10% of all HAI cases worldwide [1,4,5]. In addition, this gram-negative bacillus is reported as the second most common cause of community-acquired urinary tract infections [6]. Patients with *K. pneumoniae* infections often have a poor prognosis, and even with proper treatment, mortality rates in these cases range from 30% up to 80% [3]. Of particular concern is also the high incidence of carbapenem resistance in this species, which necessitates the use of last-resort antimicrobials such as polymyxins (i.e., colistin and polymyxin B) [4]. However, due to the increasing prescription of polymyxins, the rates of resistance to this class of antimicrobials have increased considerably, and colistin-resistant *K. pneumoniae* (CRKP), a pan-drug resistance-related phenotype, have frequently been found in health centers around the world [7,8].

Consequently, new and unconventional antimicrobials against CRKP are urgently needed. Natural or modified antimicrobial peptides (AMPs) display interesting features as potential new antibacterial agents [9,10]. AMPs are recognized components of the innate immune response of virtually all living organisms, from prokaryotes to plants and animals [11]. However, AMPs are particularly prevalent in insects, as these animals lack a cell-mediated immune response, with AMPs playing a prominent role in their immune defense [12]. These antimicrobial agents have a broad spectrum of activity against gram-positive and gram-negative bacteria, exhibit a rapid bactericidal effect, are quickly degraded in the environment (not generating residues that could contaminate the environmental microbiota), are even effective against multi-resistant bacteria, can be easily synthesized in the laboratory, and have limited potential for induction of antimicrobial resistance [9].

In this context, Jelleine-I (H-PFKISIHL-NH_2_), an octapeptide isolated from the royal jelly of honeybee (*Apis mellifera*), has shown promising antibacterial activity, including against multi-resistant pathogens such as methicillin-resistant *Staphylococcus aureus* [13], extended spectrum beta-lactamase (ESBL)-producing *Escherichia coli* [13], beta-lactam-resistant *Staphylococcus epidermidis* [14], and piperacillin-resistant *Pseudomonas aeruginosa* [13]. Furthermore, a recent study has shown that Jelleine-I has high to moderate activity against pan-drug resistant clinical isolates of *Acinetobacter baumanii* (minimum inhibitory concentration (MIC) of 8 to 16 μM), highlighting its biological activity against colistin-resistant bacteria [15]. However, there is no evidence of action of Jelleine-I on CRKP. Furthermore, it is known that the main mechanism of polymyxin resistance involves structural alterations in the outer membrane, the primary target of Jelleine-I action [16,17]. In fact, the addition of positively charged motifs, such as phosphoethanolamine (PEA) and 4-Amino-4-deoxy-L-arabinose (L-Ara4N), in the lipid A of the outer membrane and the loss of lipopolysaccharide due to mutations in the *lpxABC* operon are the main known mechanisms of polymyxin resistance in *K. pneumoniae* [16]. Thus, it is important to characterize the mode of action of Jelleine-I on the membrane of polymyxin-resistant bacteria, such as CRKP. In light of this, the purpose of the present study was to investigate the effect of Jelleine-I against CRKP in vitro and in vivo and study its activity on the cell membrane of this microorganism.

## 2. Results and Discussion

Initially, the broth microdilution assay was utilized to study the antibacterial activity of Jelleine-I against 18 clinical isolates of CRKP. As shown in Table 1, the minimum inhibitory concentration (MIC) of Jelleine-I ranged from 16 to 128 µM. The MIC_50_ and MIC_90_, defined as the MICs that inhibited 50% and 90%, respectively, of the tested microorganisms, were also determined. Jelleine-I showed MIC_50_ and MIC_90_ values of 64 µM and 83.2 µM, respectively. The antibacterial activity of this peptide was predominantly bactericidal, with the minimum bactericidal concentration (MBC) ranging from 64 to 128 µM, while MBC_50_ (concentration that kills 50% of the tested microorganisms; 128 µM) and MBC_90_ (concentration that kills 90% of the tested microorganisms; 166.4 µM) showed similar values. Related results have been reported by Jia et al. [18], who found that Jelleine-I is active against an extended-spectrum beta-lactamase-producing *K. pneumoniae* reference line (ATCC 700603) with an MIC of 64 µM and MBC of 128 µM. In their turn, other studies have reported lower MIC values; for example, Kim et al. [19] showed that Jelleine-I at 8 µM is active against *K. pneumoniae* KCTC 2242, whereas Fontana et al. [20] described a 10 µM MIC against *K. pneumoniae* ATCC 13883. Such differences of antibacterial concentrations of Jelleine-I against *K. pneumoniae* can be explained by the resistance profile of the respective isolates. Indeed, multidrug-resistant specimens were used in the present study, as well as in that by Jia et al. [18]. On the other hand, the lineages employed by Kim et al. [19] and Fontana et al. [20] are sensitive to conventional antimicrobials, resulting in expected lower MIC values.

Next, a time-kill curve assay was performed to evaluate the activity of Jelleine-I against CRKP over time. According to Figure 1, Jelleine-I at a concentration of 200 μM was able to kill all microorganisms of a CRKP bacterial suspension, which contained 10^6^ colony-forming units (CFU)/mL, after 150 min of incubation. Furthermore, a maximum microbicidal effect was also observed in suspensions treated with Jelleine-I at 100 μM after 180 min of incubation. A similar result was observed with polymyxin-resistant *Acinetobacter baumannii*, where incubation with Jelleine-I, at a concentration of 80 µM, induced maximum bactericidal effect after 180 min of incubation [15]. A rapid microbicidal effect was also observed with Jelleine-I on fungal cells; this peptide also exhibited maximum activity after 180 min of incubation against *Candida albicans* (256 µM), *Nakaseomyces glabratus* (formerly *Candida glabrata*) (128 µM), *Pichia kudriavzevii* (formerly *Candida krusei*) (128 µM), *Candida parapsilosis* (256 µM), and *Candida tropicalis* (64 µM) [23]. The fast elimination of the microbial agents by Jelleine-I is important, as it allows the infectious focus to be removed quickly, avoiding more severe consequences such as systemic infection, sepsis, prolonger antimicrobial therapy, increased hospitalization costs, superinfections, the need for invasive procedures, increased mortality, antimicrobial resistance, and disease recrudescence [24].

The action of Jelleine-I on the plasma membrane of CRKP was studied using a bacteriolytic capacity assay by measuring the release of DNA/RNA (260 nm-absorbing materials) and proteins from treated bacterial cells. Jelleine-I increased the release of 260 nm-absorbing intracellular material from CRKP cells at concentrations starting at 80 µM (Figure 2). Furthermore, the increase in protein concentration in the supernatant of a CRKP cell suspension treated with Jelleine-I revealed that this peptide indeed contributes to the extravasation of intracellular macromolecules, suggesting cell lysis. Melittin, a peptide known to lyse bacterial cells, was also able to induce release of DNA/RNA and proteins from bacterial cells, validating our experimental conditions. The increased release of cytoplasmic material is an important indication of an effect on the cell membrane, confirming numerous studies that have proposed Jelleine-I as a membranolytic peptide [15,18,25]. Castro et al. [15] showed that Jelleine-I-membrane interaction disturbs the organization of phospholipid bilayers. The authors has revealed that Jelleine-I alters the hydrodynamic diameter (Dh) and zeta potential (ζ-potential) in phospholipid vesicles synthetics that simulate bacterial membranes, beyond increasing the intracellular material release from polymyxin-resistant *Acinetobacter baummannii* [15]. Similarly, our study showed that the bacteriolytic effect of Jelleine-I is maintained even in strains with resistance to polymyxins, known for important alterations in the charge and structure of the cell membrane. These results show that resistance to polymyxins does not promote cross-resistance to Jelleine-I, suggesting the use of this AMP against Gram-negative pandrug-resistant bacteria.

Polymyxins induce disruption of the plasma membrane by displacing divalent cations (especially Ca^2+^ and Mg^2+^) bound to this cellular structure. In addition to being considered important cofactors in different microbial metabolic pathways, divalent cations play an important role in the biophysical stability of the cell membrane of microorganisms and in the osmotic resistance of the cell [26,27]. To evaluate whether the mechanism of action of Jelleine-I on the bacterial membrane is similar to that observed with polymyxins, we determined the MIC of the peptide in the presence of increasing concentrations of Ca^2+^ and Mg^2+^. Table 2 shows that the activity of Jelleine-I was not modified in the presence of divalent cations, suggesting that this compound does not act on the homeostasis of these ions. In turn, the activity of colistin is reduced in the presence of calcium and magnesium in a concentration-dependent manner (Table 2). Thus, even though the exact mode of action was not fully determined to Jelleine-I, comparative experiments with colistin showed a different mode of action, thus justifying the absence of cross-resistance between these two compounds.

Several AMPs are known to induce the formation of free radicals, especially reactive oxygen species (ROS) and reactive nitrogen species (RNS). These free radicals can disrupt microbial cell membranes, triggering oxidative stress that damages intracellular proteins and DNA, ultimately resulting in cell death [28]. To evaluate whether Jelleine-I induces oxidative damage in CRKP cells, the MIC of this peptide was determined in the presence of ascorbic acid, a potent antioxidant agent. The results, however, showed no change in MIC upon supplementation of the medium with the antioxidant, suggesting that Jelleine-I does not act by pro-oxidative pathways in CRKP. In contrast to our findings, Jia et al. [18] reported an increase in ROS concentration in *S. aureus* and *E. coli* cells treated with Jelleine-I. These differing results may be attributed to the methodology used, as Jia et al. [18] employed more sensitive fluorometric techniques for ROS measurement. However, it is important to note that an increase in ROS does not necessarily imply an antibacterial effect; verifying alterations in MIC values using pharmacological tools is crucial. Therefore, our study indicates that despite a potential increase in ROS generation, this effect likely has minimal influence on the overall antibacterial activity of Jelleine-I, as the inhibition of these agents did not alter the biological activity of the peptide.

One of the most important virulence factors of Gram-negative bacteria such as *K. pneumoniae* is their ability to form biofilms on biological tissue and inanimate surfaces. Biofilms are aggregates of microbial cells surrounded by self-produced exopolysaccharide matrices [29]. These structures exhibit enhance protection against antimicrobials, hospital sanitizers, host immune defenses, and adverse environmental conditions compared to free-living cells [30]. Therefore, the inability of conventional antimicrobial agents to target and destroy adherent bacterial biofilms is a major challenge for current antimicrobial therapies [29,30]. In this context, we investigated the effect of Jelleine-I on mature CRKP-formed biofilms in two aspects: biomass and viability of cells into biofilms. Jelleine-I was able to reduce the biomass of mature biofilm of a hypervirulent isolate of CRKP at concentrations of 80 µM (63.29 ± 12.96%; *p*-value 0.0076) and 160 µM (64.85 ± 9.06%; *p*-value 0.0108) in relation to control cells (100 ± 12.13%). Moreover, Jelleine-I also reduced the viability of CRKP cells within the biofilm at all concentrations tested (32 µM: 38.46 ± 10.33%, *p*-value = 0.0005; 80 µM: 13.72 ± 12.34%, *p*-value < 0.0001; and 160 µM: 14.91 ± 13.43%, *p*-value < 0.0001) compared to the untreated cells (100 ± 20.62%), suggesting a potent anti-biofilm effect (Figure 3). Similarly, other studies have proven the activity of Jelleine-I against biofilms of *Staphylococcus aureus* [31] and *Listeria monocytogenes* [32]. Indeed, antimicrobial peptides are known for their high activity against bacterial biofilms by inducing disruption or degradation of the membrane potential of cells embedded in the biofilm, inhibiting bacterial signaling systems by downregulating genes responsible for biofilm formation and transport of binding proteins, in addition to degrading the polysaccharide and biofilm matrix [33].

Based on the promising in vitro antibacterial activity, the Jelleine-I was investigated in a *Tenebrio molitor* larvae model of CRKP-infection. Initially, the toxicity of Jelleine-I was evaluated using a lethality assay following the administration of different doses of the peptide (1, 5, 10, 50, and 100 µg) to the larvae. As shown in Figure 4A, none of the doses studied induced significant mortality in *T. molitor* larvae during the studied period (72 h), suggesting low toxicity of Jelleine-I. In fact, previous studies have shown that Jelleine-I has low toxicity in vitro against renal cells (Vero), fibroblasts (NIH 3T3), macrophages (J774, THP_1_, and RAW264.7), cervical cells (HeLa), and murine erythrocytes, as well as in in vivo assays using Kunming mice (median lethal dose (LD_50_) > 1000 mg/kg) [20]. Next, we evaluated whether the use of Jelleine-I reduces the lethality of *T. molitor* larvae infected with a hypervirulent isolate of CRKP (10^8^ CFU). At the highest doses tested (50 µg: 35% survival after 72 h; and 100 µg: 40% survival after 72 h), there was a reduction in the lethality of infected larvae when compared to the untreated group (0% survival after 24 h; *p*-value < 0.05) (Figure 4B). Furthermore, the general clinical score of infected larvae was higher in the groups treated with doses of 50 µg (3 (0–8)) and 100 µg (1 (0–7)) of Jelleine-I, when compared to larvae treated with saline (1 (0–4)) (Figure 4C). Similar results were observed by Jia et al. [31], who showed that neutropenic mice infected with *E. coli* treated with Jelleine-I (20 mg/Kg) had a survival rate of 25% after 10 days. Furthermore, Jia et al. [23] have shown that mice infected with *C. albicans* and treated with Jelleine-I (10 mg/Kg) daily for seven days have a 40% lethality after 14 days, which is significantly lower compared to untreated animals (100% lethality). In addition, in an invertebrate model, using *Galleria mellonella* larvae, Jelleine-I also caused significant protection against *Listeria monocytogenes*-induced lethality by increasing survival rates by 10%, 20%, and 30% at 10 mg/kg, 20 mg/kg, and 40 mg/kg, respectively [32].

## 3. Conclusions

The results show that Jelleine-I exhibits promising in vitro and in vivo activity against CRKP. Jelleine-I acts on planktonic cells and on biofilms of pandrug-resistant *K. pneumoniae*, exhibiting rapid bactericidal effect in the time-kill curve assay. Although the antimicrobial effect of Jelleine-I involves damage to the bacterial membrane, this mechanism of action differs from that observed with polymyxins, which act by displacing divalent cations bound to the plasma membrane. This explains the absence of cross-resistance between Jelleine-I and polymyxins in *K. pneumoniae*. Furthermore, this peptide proved to be safe and effective in in vivo assays using *T. molitor* larvae, being able to reduce the lethality of animals infected with a hypervirulent strain of CRKP at doses without detectable toxicity. These results also highlights the use of *T. molitor* larvae as a viable model to study the virulence and treatment of *K. pneumoniae*, which could be explored as a replacement for vertebrate animal models. In conclusion, this study highlights Jelleine-I as a promising prototype for the development of a novel antimicrobial against pandrug-resistant *K. pneumoniae*.

## 4. Materials and Methods

### 4.1. Reagents

Colistin, meropenem (Inlab^TM^, São Paulo, SP, Brazil), crystal violet, glacial acetic acid, methanol, sodium chloride, magnesium chloride, calcium chloride, glucose (Synth^TM^, São Paulo, SP, Brazil), 3-(4,5-dimethylthiazol-2-yl)-2,5-diphenyltetrazolium bromide (MTT), Coomassie brilliant blue G 250, and bovine serum albumin (Sigma-Aldrich^TM^, Frankfurt, HE, Germany) were purchased from commercial suppliers and used without further purification because the purity was greater than 99% in all cases. Mueller–Hinton broth (MHB), Muelle–-Hinton agar (MHA), and MacConkey agar were purchased from Kasvi^TM^ (São José do Pinhais, PR, Brazil). Jelleine-I was obtained through solid-phase synthesis and subsequently purified and characterized in the Peptide Synthesis and Structure Laboratory (LASEP/UFMG), exactly as described previously [15]. After purification, a purity of 97.48% was obtained.

### 4.2. Microorganisms

In this study, were employed eighteen clinical isolates of colistin-resistant *Klebsiella pneumoniae* (CRKP) recovered from patients admitted in a tertiary hospital in Belo Horizonte (Minas Gerais, Brazil). All isolates were identified using biochemical-morphological tolls by an automated BD Phoenix^TM^ system (BD^TM^, New York, NY, USA). Resistance to colistin was initially evaluated by the growth of isolates in MHB supplemented with colistin at a concentration of 4 µg/mL, as recommended by the Brazilian Committee on Antimicrobial Susceptibility Testing (BrCAST). Next, the resistance to colistin among the isolates that grew in the medium supplemented with this antimicrobial was confirmed by determination of minimum inhibitory concentration (MIC) second to the Clinical Laboratory and Standard Institute (CLSI) [34]. In addition, two reference lineages from the American Type Culture Collection^TM^ (ATCC) (*K. pneumoniae* ATCC 700603 and *K. pneumoniae* ATCC 43816) were kindly provided by the João XXIII hospital (Belo Horizonte, MG, Brazil) and included in this study as internal controls.

### 4.3. Antibacterial Activity

*Inoculum preparation:* The bacterial inoculum was standardized according to the CLSI rules [34]. Two to three isolated colonies, which were collected from a 24 h CRKP culture realized on MHA, were suspended in a sterile saline solution (0.85% NaCl; 10 mL) with the aid of a bacteriological loop. Next, the resulting suspension was adjusted to an optical density (OD) of 0.08–0.13 at 625 nm (Nova Instruments^TM^, Sao Paulo, SP, Brazil), which is equivalent to the McFarland 0.5 scale (10^8^ colony-forming units (CFU)/mL). Finally, 50 μL of the bacterial suspension prepared in saline was transferred to MHB (10 mL) to produce a working inoculum at 10^6^ CFU/mL.

*Minimum inhibitory concentration (MIC):* Bacteriostatic activity was assessed by determining the MIC according to the CLSI guidelines [34], with minor modifications [35]. One hundred microliters (100 μL) of a working inoculum of CRKP (10^6^ CFU/mL) were added to microplates containing 100 μL of a twofold serial dilution (1–128 µM) of Jelleine-I or colistin in MHB. The concentration range chosen for the tests with Jelleine-I was determined based on previous studies by Castro et al. (2025), who evaluated the antibacterial effect of this peptide against pandrug resistant *Acinetobacter baumannii* [15]. The plates were then incubated at 35 ± 2 °C for 18 h, and the MIC was defined as the lowest concentration of the compounds that visibly inhibited bacterial growth. Each plate included wells with 200 µL of culture medium as a sterility control and wells with 100 µL of culture medium plus 100 µL of inoculum as a microorganism viability control.

*Minimum bactericidal concentration (MBC):* Bactericidal activity was assessed by transferring 100 µL from the wells showing no visible growth in the MIC assay to MHA plates, followed by incubation at 35 ± 2 °C for 24 h [35]. The MBC was defined as the lowest concentration of Jelleine-I that resulted in complete inhibition of bacterial growth.

### 4.4. Time-Kill Curve

A pre-inoculum of CRKP (isolate 699) at a concentration of 10^8^ CFU/mL was prepared in sterile saline (0.85% NaCl). Subsequently, 50 µL of pre-inoculum were added to test tubes containing MHB (10 mL) with Jelleine-I at different concentrations (50, 100, or 200 µM). The tubes were incubated at 35 ± 2 °C, and, in specific time points (0, 30, 60, 90, 120, 150, and 180 min), samples were serially diluted (10^−1^ to 10^−6^) in sterile saline (0.85% NaCl) and plated onto MacConkey agar. The plates were then incubated at 35 ± 2 °C for 24 h, and bacterial counts were determined and expressed as CFU/mL. Untreated cells served as the negative control [35].

### 4.5. Release of Bacterial DNA/RNA

The release of intracellular components (DNA/RNA), indicated by absorbance at 260 nm, was quantified [35]. Bacterial suspensions of CRKP in 0.85% saline (10^8^ CFU/mL) were treated with Jelleine-I at concentrations of 32, 80, or 160 µM. After incubation, 100 µL of each suspension were centrifuged (1000× *g* for 25 min at 4 °C), and the absorbance of the resulting supernatant was measured at the wavelength of 260 nm (Shimadzu™, Tokyo, OS, Japan). Finally, the results were plotted as OD_620_ versus time (hours). Melittin (100 µM) was used as a positive control.

### 4.6. Release of Protein

Protein release was studied by the Bradford method [36]. After exposure of a CRKP suspension at 10^6^ CFU/mL prepared in saline (0.85% NaCl) to different concentrations of Jelleine-I (32, 80, or 160 µM) for 24 h, aliquots (100 µL) were collected and centrifuged (1000× *g* for 25 min at 4 °C). The supernatant was mixed with Coomassie brilliant blue G 250, and, after 2 min of reaction, the blue chromogen was measured spectrophotometrically at 595 nm (Nova Instruments^TM^, Sao Paulo, SP, Brazil). The protein concentration in the supernatant was determined with the aid of the straight-line equation obtained from a standard curve constructed with bovine serum albumin (2.5–100 µg/mL). Untreated cells were included as a negative control, and cells exposed to melittin, an agent known to lyse bacterial cells, were used as a positive control.

### 4.7. Binding to Divalent Cations

The ability of Jelleine-I to displace membrane-bound divalent cations (Ca^2+^ and Mg^2+^) from the membrane of CRKP was assessed by determining the MIC after supplementation of the medium with different concentrations of calcium chloride (20, 50, and 100 µM) and magnesium chloride (20, 50, and 100 µM) [37]. Increases in MIC from two dilutions in the supplemented media in relation to the control medium were considered suggestive of binding to divalent cations. Colistin, an antimicrobial from polymyxin class known to act by binding to cations present in the bacterial cell membrane, was used as a positive control.

### 4.8. Induction of Oxidative Stress

Oxidative damage induced by Jelleine-I was evaluated by determining the minimum inhibitory concentration (MIC) in MHB supplemented with ascorbic acid (100 µg/mL), a well-known antioxidant [38]. An increase in the MIC by at least twofold dilutions following the addition of ascorbic acid was considered indicative of oxidative damage.

### 4.9. Anti-Biofilm Assay

A CRKP inoculum (1 × 10^6^ CFU/mL; isolate 699) was incubated in MHB supplemented with 1 mM glucose at 35 ± 2 °C for 24 h to allow biofilm adhesion and formation. After incubation, the medium was discarded, and wells were washed with sterile saline (0.85% NaCl; three washes). Biofilms were then treated with Jelleine-I at concentrations of 32, 80, or 160 µM, followed by a second 24 h incubation at 35 ± 2 °C. Subsequently, the medium was removed, wells were washed again with sterile saline (three times), and the biofilms were fixed with methanol for 5 min at 37 °C. Fixed biofilms were stained with 0.1% (*w*/*v*) crystal violet for 30 min at room temperature, washed with saline (three times), air-dried, and solubilized with 30% glacial acetic acid. Absorbance was measured at 595 nm using a spectrophotometer (Bio-Tek Instruments™, Winooski, VT, USA), and results were expressed as the percentage of biofilm biomass relative to untreated control wells (set as 100%) [39]. In parallel, biofilm cell viability was evaluated using the MTT assay, as described previously [40].

### 4.10. In Vivo Assay

Invertebrates can provide a valuable alternative to traditional vertebrate animal models for studying the effect of antimicrobial compounds. In this sense, several studies have traditionally used *Galleria mellonella* larvae as a model to evaluate the antimicrobial properties of various compounds [41]. However, access and availability of *G. mellonella* larvae is limited in several regions, such as Brazil. To overcome this limitation, some studies have used *Tenebrio molitor* (Coleoptera: Tenebrionidae) larvae as an alternative, since the widespread use of this species as a nutritional factor for farm animals makes its access significantly easier [42,43,44]. Therefore, we chose to employ a *T. molitor* larval infection model to study the antibacterial activity of Jelleine-I against a hypervirulent strain of colistin-resistant *Klebsiella pneumoniae* (isolate 699).

Larvae of *Tenebrio molitor* (100–120 mg and 2 cm) were purchased from a regional commerce (Alvorada dos Pássaros^TM^, Belo Horizonte, Mg, Brazil). All animals were maintained in plastic containers protected from light and filled with a commercial mixture of seeds and grains. Groups of ten larvae, randomly selected, were separated and placed in petri dishes. In the toxicity test, eight groups (*n* = 10) were prepared, and the larvae of each group were inoculated with a micro-syringe (Hamilton^TM^, Reno, NV, USA) containing 5 μL of solutions containing different doses of Jellein-I (1, 5, 10, 50, and 100 μg). A group treated with saline (NaCl 0.85%; 5 μL) and a control group of non-injected larvae were included as a negative control. In addition, a wound control group, in which the needle was inserted without the introduction of any vehicle, was used to evaluate the effect of trauma on larval survival. After the respective treatments, all groups were incubated at 37 °C for 72 h. Larval mortality was monitored daily, and individuals were considered dead if they were immobile, unresponsive to touch, or exhibited melanization, as described by Andrade-Oliveira et al. [42]. In the efficacy assay, each larva was injected with 5 μL of a hypervirulent colistin-resistant *K. pneumoniae* suspension (10^8^ CFU) using a microsyringe. Two hours post-infection, larvae were treated with Jelleine-I at doses of 10, 50, or 100 μg, or with sterile 0.85% saline (control), then placed in Petri dishes and incubated at 37 °C for 72 h to support bacterial proliferation. Mortality and morbidity were assessed daily. Larval health was evaluated over the 72 h period based on three criteria, activity, melanization, and survival, using the health index scoring system anteriorly described [42].

### 4.11. Statistical Analysis

Data normality was assessed using the Shapiro–Wilk test. Data with normal distribution were represented as mean ± standard deviation, and data with non-Gaussian distribution were represented as median and interquartile range (P_25_–P_75_). One-way analysis of variance (ANOVA) followed by Tukey’s post-test were used to compare differences between data with normal distribution (release of 260 nm-absorbing material, release of proteins, biomass of biofilm, and viability of biofilm). Kruskall–Wallis followed by Dunn’s post-test was used to compare differences between data with non-normal distribution (clinical score). The lethality curves of *T. molitor* larvae were constructed using the Kaplan–Meier estimator, and the comparison between groups was performed by the log-rank test. All statistical analyses were assessed using GraphPad Prism 5.03 (GraphPad Software Inc. ^TM^, LaJolla, CA, USA), and *p* values < 0.05 were considered statistically significant.

## Figures and Tables

**Figure 1 toxins-17-00325-f001:**
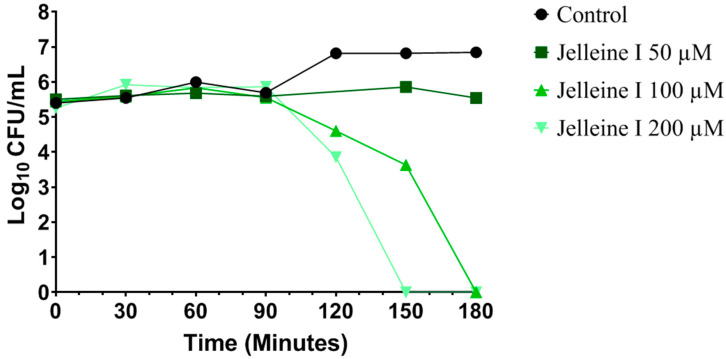
Time-kill curve with multiple concentrations of Jelleine-I (50, 100, and 200 µM) against a colistin-resistant *Klebsiella pneumoniae* lineage (isolate 699). The plot shows the number of logarithmic colony-forming units per milliliter (Log_10_ CFU/mL). Bacterial cells untreated were used as negative control (black circles).

**Figure 2 toxins-17-00325-f002:**
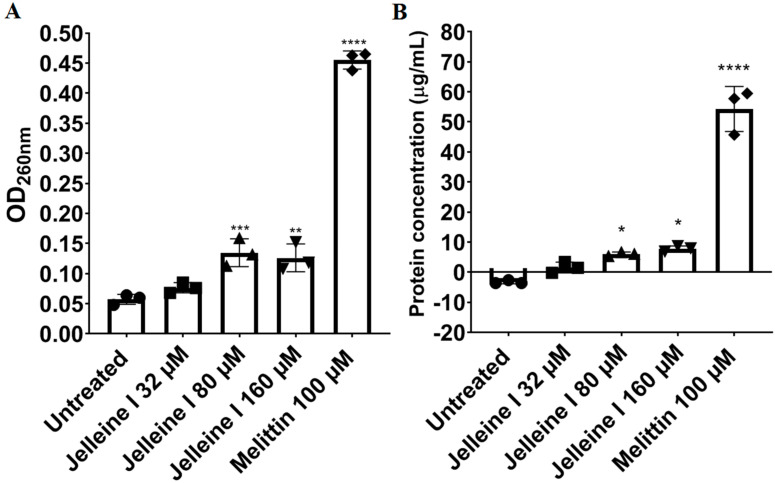
Determination of bacteriolysis through of assay of overflow of 260 nm-absorbing materials (DNA/RNA) (**A**) and through a test of release of proteins (**B**). The experiment was done in triplicate for statistical significance. One asterisk (*) indicates statistically different compared to the control with 0.05 < *p*-value < 0.01. Two asterisks (**) indicate statistically different compared to the control with 0.01 < *p*-value < 0.001. Three asterisks (***) indicate statistically different compared to the control with 0.001 < *p*-value < 0.0001. Four asterisks (****) indicate statistically different compared to the control with *p*-value < 0.0001. The results were analyzed by one-way variance analysis (ANOVA) with Dunnett post-hoc.

**Figure 3 toxins-17-00325-f003:**
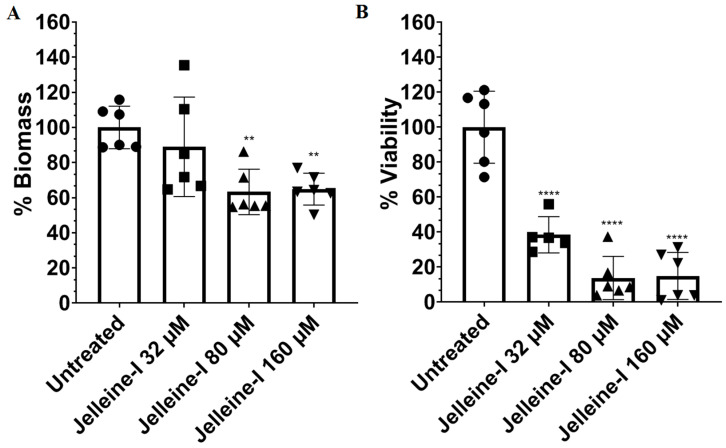
Effect of different concentrations of Jelleine-I (32, 80, and 160 µM) on biomass (**A**) and viability (**B**) of biofilms formed by a hypervirulent colistin-resistant *Klebsiella pneumoniae* (isolate 699). The experiment was done in sextuplicate for statistical significance. Two asterisks (**) indicate statistically different compared to the control with 0.01 < *p*-value < 0.001. Four asterisks (****) indicate statistically different compared to the control with *p*-value < 0.0001. The results were analyzed by one-way variance analysis (ANOVA) with Dunnett post-hoc.

**Figure 4 toxins-17-00325-f004:**
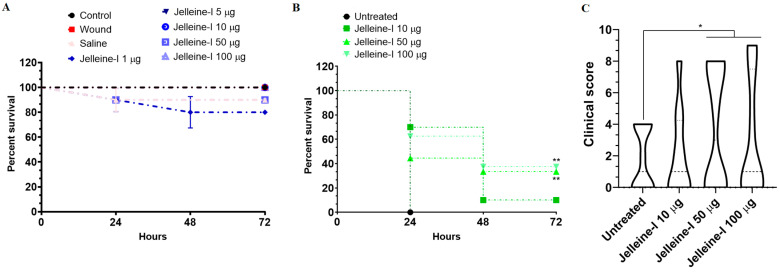
Survival curves and clinical score of *Tenebrio molitor* larvae. (**A**) Survival of larvae after administration of different doses of Jelleine-I (1, 5, 10, 50, and 100 µg) to assess the in vivo toxicity of the peptide. (**B**) Survival curve of larvae infected with a hypervirulent clinical isolate of colistin-resistant *Klebsiella pneumoniae* (CRKP; isolate 699). Groups (n = 10) infected with CRKP and treated with Jelleine-I (10, 50, and 100 µg) after 2 h post-infection. Larvae (n = 10) infected and injected with sterile saline (0.9% NaCl) were used as the negative control. (**C**) Clinical score of larvae infected and treated with Jelleine-I (10, 50, and 100 µg) or saline. One asterisk (*) indicates statistically different compared to the control with 0.05 < *p* < 0.01. The lethality curves were constructed using the Kaplan–Meier estimator and statistically analyzed using the log-rank test. The clinical score results were analyzed using the Kruskal–Wallis test with Dunn’s post-test. The results are expressed as median and interquartile range (P_25_–P_75_).

**Table 1 toxins-17-00325-t001:** Values of minimum inhibitory concentration (MIC) and minimum bactericidal concentration (MBC) of Jelleine-I peptide against clinical isolates of colistin-resistant *Klebsiella pneumoniae*.

Microorganisms	Origin	Carbapenem Resistance ^a^	Hypermucoviscity Phenotype ^b^	Jelleine-I (µM)		Colistin (µg/mL)	
				MIC	MBC	MIC	Profile ^c^
*K. pneumoniae* ATCC 700603	Urine	No	No	32	128	0.25	Sensible
*K. pneumoniae* ATCC 43816	Secretion	No	Yes	32	128	0.5	Sensible
*K. pneumoniae* 17	Secretion	Yes	Yes	64	64	32	Resistant
*K. pneumoniae* 27	Secretion	Yes	No	64	>128	64	Resistant
*K. pneumoniae* 68	Secretion	Yes	Yes	16	32	8	Resistant
*K. pneumoniae* 76	Secretion	Yes	Yes	64	64	32	Resistant
*K. pneumoniae* 104	Urine	Yes	Yes	64	64	64	Resistant
*K. pneumoniae* 106	Urine	Yes	No	128	128	>128	Resistant
*K. pneumoniae* 110	Urine	Yes	Yes	64	64	32	Resistant
*K. pneumoniae* 135	Blood	Yes	Yes	64	128	64	Resistant
*K. pneumoniae* 139	Blood	Yes	Yes	64	128	64	Resistant
*K. pneumoniae* 169	Urine	Yes	No	64	64	32	Resistant
*K. pneumoniae* 242	Secretion	Yes	No	128	128	32	Resistant
*K. pneumoniae* 260	Secretion	Yes	No	64	64	16	Resistant
*K. pneumoniae* 289	Blood	Yes	No	64	128	32	Resistant
*K. pneumoniae* 395	Blood	Yes	No	64	128	32	Resistant
*K. pneumoniae* 627	Secretion	Yes	No	64	>128	32	Resistant
*K. pneumoniae* 642	Secretion	Yes	No	64	64	32	Resistant
*K. pneumoniae* 689	Blood	Yes	Yes	64	128	32	Resistant
*K. pneumoniae* 699	Blood	Yes	Yes	64	128	128	Resistant

^a^ Resistance to carbapenems was confirmed by the presence of visible growth (turbidity) in culture medium containing 8 µg/mL of meropenem [21]. ^b^ The method of the International *Klebsiella pneumoniae* Study Group was adopted [22]. ^c^ The resistance profile was judged according to the cutoff points established in the Brazilian Committee on Antimicrobial Susceptibility Testing (BrCAST), because the bacterial isolates included in this study were recovered from a tertiary hospital localized in southeastern Brazil. Thus, we consider colistin resistance to be assumed when the MIC is >2 µg/mL [21].

**Table 2 toxins-17-00325-t002:** Influence of divalent cations (Ca^2+^ and Mg^2+^) and ascorbic acid (100 µg/mL) on the values of minimum inhibitory concentration (MIC; µg/mL) of Jelleine-I or colistin against a hypervirulent clinical isolate of colistin-resistant *Klebsiella pneumoniae* (isolate 699).

	Control	Ca^2+^			Mg^2+^			Ascorbic Acid 100 µg/mL
		20 µM	50 µM	100 µM	20 µM	50 µM	100 µM	
Jelleine-I	16	16	16	16	16	16	16	16
Colistin	8	8	16	>64	8	16	>64	8

## Data Availability

The original contributions presented in this study are included in the article. Further inquiries can be directed to the corresponding author.

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
