# Peer review of "Antibacterial Activity of Jelleine-I, a Peptide Isolated from Royal Jelly of Apis mellifera, Against Colistin-Resistant Klebsiella pneumoniae"

_toxins, 2025, doi:10.3390/toxins17070325_

Round 1
Reviewer 1 Report
Comments and Suggestions for Authors
The authors' studies on the "Antibacterial activity of Jelleine-I, a peptide isolated from royal jelly of Apis mellifera, against colistin-resistant Klebsiella pneumoniae" are very well presented and scientifically sound. The paper was easy to read and follow through and I think the inclusion of larvae made a significant difference to the overall merit of the paper.
My only concern is around the interchange of μg/mL and μM. Both the controls and the peptide need to be in the same way to make it clear to compare. On that note, am I missing something or was there no control (another active antibacterial agent) in the biofilm studies?
Are there any larvae pics and anything else to include as supp info?
Was the peptide synthesized or commercially bought? In any case, as always we need the purity of the peptide and if it was synthesized the method of synthesis etc.
Comments on the Quality of English LanguageIn the introduction and abstract, you mention that its mode of action is different than Polymyxin but you are not stating the actual mode of action. If you haven't concluded in the mode of action rephrase to make it clear. Something among the lines of: "...even though the exact mode of action was not fully determined, comparative experiments with Polymyxin showed a different mode of action"
Author Response
Overall comment: The authors' studies on the "Antibacterial activity of Jelleine-I, a peptide isolated from royal jelly of Apis mellifera, against colistin-resistant Klebsiella pneumoniae" are very well presented and scientifically sound. The paper was easy to read and follow through and I think the inclusion of larvae made a significant difference to the overall merit of the paper.
Answer: We thank the reviewer for his comments and corrections, which certainly contributed to improving the quality of our manuscript. We have adapted the manuscript according to the recommendations presented.
Question #1: My only concern is around the interchange of μg/mL and μM. Both the controls and the peptide need to be in the same way to make it clear to compare. On that note, am I missing something or was there no control (another active antibacterial agent) in the biofilm studies?
Answer: We thank the reviewer for the comment. We found that we mistakenly reported the results for Jelleine-I in terms of μg/mL in the manuscript, although it is correctly represented in terms of μM in Table 1. This information was corrected in the text. Regarding the colistin unit, it needs to be in μg/mL because we use this data to confirm resistance to polymyxins. To do this, we need to compare the result with the cutoff points defined in BrCAST, which are all in terms of mg/L, which is interchangeable with μg/mL, but not with μM. We did not use a positive control because, as this is a pan-resistant bacterium, we do not have any antimicrobial available that is effective against this species. Therefore, our results highlight the relevance of Jelleine-I activity in this context.
Question #2: Are there any larvae pics and anything else to include as supp info?
Answer: We did not take photographs of the larvae. However, we will consider these data in future studies, so we appreciate the reviewer's point.
Question #3: Was the peptide synthesized or commercially bought? In any case, as always we need the purity of the peptide and if it was synthesized the method of synthesis etc.
Answer: We thank the reviewer for the comment. Information about the origin of the peptide used has been added in the "Reagents" section (Material and methods).
Question #4: In the introduction and abstract, you mention that its mode of action is different than Polymyxin but you are not stating the actual mode of action. If you haven't concluded in the mode of action rephrase to make it clear. Something among the lines of: "...even though the exact mode of action was not fully determined, comparative experiments with Polymyxin showed a different mode of action"
Answer: The correction was made as suggested by the reviewer.
Reviewer 2 Report
Comments and Suggestions for Authors
Antibacterial activity of Jelleine-I, a peptide isolated from royal jelly of Apis mellifera, against colistin-resistant Klebsiella pneumonia
Abstract section: Please rewrite this section. Divide this section like what was the purpose? Which methodology was used? What was the observation during the experimental work? Finally what kind of output your study provides?
Section 2 Results and Discussion
This section is poorly written. Authors should compare their results and observation in a clear way. Try to add the specific reasons behind such kind of observations. Use observations from the latest scientific findings.
Section 3. Conclusions
Conclusion is too short. Elaborate this section.
- Materials and Methods
4.1. Reagents : Purity level of chemical reagents are not mentioned. Please provide the missing detail
Author Response
Overall comment: Antibacterial activity of Jelleine-I, a peptide isolated from royal jelly of Apis mellifera, against colistin-resistant Klebsiella pneumonia
Answer: We thank the reviewer for his comments and corrections. We have adapted the manuscript according to the recommendations presented.
Question #1: Abstract section: Please rewrite this section. Divide this section like what was the purpose? Which methodology was used? What was the observation during the experimental work? Finally what kind of output your study provides?
Answer: The correction was made as suggested by the reviewer.
Question #2: Section 2 Results and Discussion
This section is poorly written. Authors should compare their results and observation in a clear way. Try to add the specific reasons behind such kind of observations. Use observations from the latest scientific findings.
Answer: The correction was made as suggested by the reviewer.
Question #3: Section 3. Conclusions
Conclusion is too short. Elaborate this section.
Answer: The correction was made as suggested by the reviewer.
Question #4: Materials and Methods
4.1. Reagents : Purity level of chemical reagents are not mentioned. Please provide the missing detail
Answer: The correction was made as suggested by the reviewer.
Reviewer 3 Report
Comments and Suggestions for Authors
The manuscript titled 'Antibacterial Activity of Jelleine-I, a Peptide Isolated from Royal Jelly of Apis mellifera, against Colistin-Resistant Klebsiella pneumoniae' is well-written and provides valuable insights. However, to meet the standards for final publication, the following improvements are recommended:
Line 16: Verify the range of values (32 to >128 µM) for accuracy and consistency with the data presented.
Line 17: Avoid abbreviations in the abstract, such as CFU. Use the full term (e.g., Colony-Forming Units) at first mention, followed by the abbreviation in parentheses if needed.
Line 70: Confirm the full form of MIC is correctly stated.
Table 1: Correct the spelling of “Origen” to “Origin.”
Table 1: Ensure unit consistency for Jelleine-I MIC and MBC values. The table uses µM, while the text uses µg/mL. Standardize units throughout the manuscript for clarity.
Line 134: Replace “several concentrations” with “multiple concentrations” or “a range of concentrations” for precision.
Line 136: Insert a space in “Log106CFU/mL” to read “Log106 CFU/mL” for proper formatting.
Line 148: Add the missing reference number for Castro et al. to ensure proper citation.
Figure 2B: Label the Y-axis as “Protein Concentration” for clarity and specificity.
Table 2: Clarify the procedure for determining ascorbic acid concentration, as it is not described in the Materials and Methods section.
Figure 3: Add a space between “%” and “Viability” in the axis title (e.g., “% Viability”) for correct formatting.
Line 324: Justify the selection of the Jelleine-I concentration range (1–128 µM). Provide the rationale or experimental basis for this choice.
Line 361: Correct the formatting of “106CFU/mL” to “106 CFU/mL” for consistency and clarity.
Rationale for Tenebrio molitor: Explain the rationale for selecting Tenebrio molitor as the model organism for testing Jelleine-I, including its relevance to the study’s objectives.
Line 510: Correct the typographical error (please specify the error if known, e.g., spelling, punctuation).
Author Response
Overall comments: The manuscript titled 'Antibacterial Activity of Jelleine-I, a Peptide Isolated from Royal Jelly of Apis mellifera, against Colistin-Resistant Klebsiella pneumoniae' is well-written and provides valuable insights. However, to meet the standards for final publication, the following improvements are recommended
Answer: We thank the reviewer for his comments and corrections, which certainly contributed to improving the quality of our manuscript. We have adapted the manuscript according to the recommendations presented.
Question #1: Line 16: Verify the range of values (32 to >128 µM) for accuracy and consistency with the data presented.
Answer: The correction was made as suggested by the reviewer.
Question #2: Line 17: Avoid abbreviations in the abstract, such as CFU. Use the full term (e.g., Colony-Forming Units) at first mention, followed by the abbreviation in parentheses if needed.
Answer: The correction was made as suggested by the reviewer.
Question #3: Line 70: Confirm the full form of MIC is correctly stated.
Answer: The correction was made as suggested by the reviewer.
Question #4: Table 1: Correct the spelling of “Origen” to “Origin.”
Answer: The correction was made as suggested by the reviewer.
Question #5: Table 1: Ensure unit consistency for Jelleine-I MIC and MBC values. The table uses µM, while the text uses µg/mL. Standardize units throughout the manuscript for clarity.
Answer: The correction was made as suggested by the reviewer.
Question #6: Line 134: Replace “several concentrations” with “multiple concentrations” or “a range of concentrations” for precision.
Answer: The correction was made as suggested by the reviewer.
Question #7: Line 136: Insert a space in “Log106CFU/mL” to read “Log106 CFU/mL” for proper formatting.
Answer: The correction was made as suggested by the reviewer.
Question #8: Line 148: Add the missing reference number for Castro et al. to ensure proper citation.
Answer: The correction was made as suggested by the reviewer.
Question #9: Figure 2B: Label the Y-axis as “Protein Concentration” for clarity and specificity.
Answer: The correction was made as suggested by the reviewer.
Question #10: Table 2: Clarify the procedure for determining ascorbic acid concentration, as it is not described in the Materials and Methods section.
Answer: We forgot to add the methodology for the ascorbic acid experiments. So, we added the item "Induction of oxidative stress" to the Material and Methods section, which details the ascorbic acid methodology. In addition, the reference on which we based the standardization of this method was also cited.
Question #11: Figure 3: Add a space between “%” and “Viability” in the axis title (e.g., “% Viability”) for correct formatting.
Answer: The correction was made as suggested by the reviewer.
Question #12: Line 324: Justify the selection of the Jelleine-I concentration range (1–128 µM). Provide the rationale or experimental basis for this choice.
Answer: The correction was made as suggested by the reviewer in section “Antibacterial activity” (Material and Methods).
Question #13: Line 361: Correct the formatting of “106CFU/mL” to “106 CFU/mL” for consistency and clarity.
Answer: The correction was made as suggested by the reviewer.
Question #14: Rationale for Tenebrio molitor: Explain the rationale for selecting Tenebrio molitor as the model organism for testing Jelleine-I, including its relevance to the study’s objectives.
Answer: The information was added as suggested by the reviewer in “In vivo assay” section (Material and Methods).
Question #15: Line 510: Correct the typographical error (please specify the error if known, e.g., spelling, punctuation).
Answer: The correction was made as suggested by the reviewer.
Round 2
Reviewer 3 Report
Comments and Suggestions for Authors
Authors have addressed all comments.